# Learning Online Data Association

## Abstract

When an agent interacts with a complex environment, it receives a stream of percepts in which it may detect entities, such as objects or people. To build up a coherent, low-variance estimate of the underlying state, it is necessary to fuse information from multiple detections over time. To do this fusion, the agent must decide which detections to associate with one another. We address this data-association problem in the setting of an online filter, in which each observation is processed by aggregating into an existing object hypothesis. Classic methods with strong probabilistic foundations exist, but they are computationally expensive and require models that can be difficult to acquire. In this work, we use the deep-learning tools of sparse attention and representation learning to learn a machine that processes a stream of detections and outputs a set of hypotheses about objects in the world. We evaluate this approach on simple clustering problems, problems with dynamics, and complex image-based domains. We find that it generalizes well from short to long observation sequences and from a few to many hypotheses, outperforming other learning approaches and classical non-learning methods.

## 1 Introduction

Consider a robot operating in a household, making observations of multiple objects as it moves around over the course of days or weeks. The objects may be moved by the inhabitants, even when the robot is not observing them, and we expect the robot to be able to find any of the objects when requested. We will call this type of problem *entity monitoring*. It occurs in many applications, but we are particularly motivated by the robotics applications where the observations are very high dimensional, such as images. Such systems need to perform online *data association*, determining which individual objects generated each observation, and *state estimation*, aggregating the observations of each individual object to obtain a representation that is lower variance and more complete than any individual observation. This problem can be addressed by an online recursive *filtering* algorithm that receives a stream of object detections as input and generates, after each input observation, a set of hypotheses corresponding to the actual objects observed by the agent.

When observations are closely spaced in time, the entity monitoring problem becomes one of *tracking* and it can be constrained by knowledge of the object dynamics. In many important domains, such as the household domain, temporally dense observations are not available, and so it is important to have systems that do not depend on continuous visual tracking.

A classical solution to the entity monitoring problem, developed for the tracking case but extensible to other dynamic settings, is a *data association filter* (DAF) (the tutorial of Bar-Shalom et al. (2009) provides a good introduction). A Bayes-optimal solution to this problem can be formulated, but it requires representing a number of possible hypotheses that grows exponentially with the number of observations. A much more practical, though much less robust, approach is a maximum likelihood DAF (ML-DAF), which commits, on each step, to a maximum likelihood data association: the algorithm maintains a set of object hypotheses, one for each object (generally starting with the empty set) and for each observation it decides to either: (a) associate the observation with an existing object hypothesis and perform a Bayesian update on that hypothesis with the new data, (b) start a new object hypothesis based on this observation, or (c) discard the observation as noise.

The engineering approach to constructing a ML-DAF requires many design choices, including the specification of a latent state space for object hypotheses, a generative model relating observations to objects, and thresholds or other decision rules for choosing, for a new observation, whether to associate it with an existing hypothesis, use it to start a new hypothesis, or discard it. In any particular application, the engineer must tune all of these models and parameters to build a DAF that performs well. This is a time-consuming process that must be repeated for each new application, and it is effectively impossible to do by hand when observations and hypothesis are high dimensional.

A special case of entity monitoring is one in which the objects' state is static, and does not change over time. In this case, a classical solution is online (robust) clustering. Online clustering algorithms perform data association (cluster assignment) and state estimation (computing a cluster center).

In this paper we explore DAFs for dynamic entity monitoring and as online clustering methods for static entity monitoring. Although it is possible to train an unstructured RNN to solve these problems, we believe that building in some aspects of the structure of the DAF will allow faster learning with less data and allow the system to address problems with a longer horizon. We begin by briefly surveying the related literature, particularly focused on learning-based approaches. We then describe a neural-network architecture that uses self-attention as a mechanism for data association, and demonstrate its effectiveness in several illustrative problems. We find that it outperforms a raw RNN as well as domain-agnostic online clustering algorithms, and competitively with batch clustering strategies that can see all available data at once and with state-of-the-art DAFs for tracking with hand-built dynamics and observation models. Finally, we illustrate its application to problems with images as observations in which both data association and the use of an appropriate latent space are critical.

## 2 Related Work

**Online clustering methods**    The typical setting for clustering problems is *batch*, where all the data is presented to the algorithm at once, and it computes either an assignment of data points to clusters or a set of cluster means, centers, or distributions. We are interested in the *online* setting, with observations arriving sequentially and a cumulative set of hypotheses output after each observation One of the most basic online clustering methods is *vector quantization*, articulated originally by Gray (1984) and understood as a stochastic gradient method by Kohonen (1995). It initializes cluster centers at random and assigns each new observation to the closest cluster center, and updates that center to be closer to the observation. Methods with stronger theoretical guaranteees, and those that handle unknown numbers of clusters have also been developed. Charikar et al. (2004) formulate the problem of online clustering, and present several algorithms with provable properties. Liberty et al. (2016) explore online clustering in terms of the facility allocation problem, using a probabilistic threshold to allocate new clusters in data. Choromanska & Monteleoni (2012) formulate online clustering as a mixture of separate expert clustering algorithms. Online clustering has also been studied in the data stream community. Silva et al. (2013) provide a survey of clustering approachs for data streams, which sometimes allow multiple passes through the data. Interesting variations construct a core-set of points to be clustered Ackermann et al. (2012) and to maintain balanced trees online Kobren et al. (2017).

**Learning for clustering**    There is a great deal of work using deep-learning methods to find latent spaces for clustering complex objects, particularly images. Min et al. (2018) provide an excellent survey, including methods with auto-encoders, GANs, and VAEs. Relevant to our approach are *amortized inference* methods, including *set transformers* (Lee et al., 2018) and its specialization to *deep amortized clustering* (Lee et al., 2019), in which a neural network is trained to map directly from data to be clustered into cluster assignments or centers. A related method is *neural clustering processes* (Pakman et al., 2019), which includes an online version, and focuses on generating samples from a distribution on cluster assignments, including an unknown number of clusters.

**Dynamic domains**    In the setting when the underlying entities have dynamics, such as airplanes observed via radar, a large number of DAFs have been developed. The most basic filter, for the case of a single entity and no data association problem, is the Kalman filter (Welch & Bishop, 2006). In the presence of data-association uncertainty the Kalman filter can be extended by considering assignments of observations to

multiple existing hypotheses under the multiple hypothesis tracking (MHT) filter. A more practical approach that does not suffer from the combinatorial explosion of the MHT is the joint probabilistic data association (JPDA) filter, which keeps only one hypothesis but explicitly reasons about the most likely assignment of observations to hypotheses. Bar-Shalom et al. (2009) provides a detailed overview and comparison of these approaches, all of which require hand-tuned transition and observation models.

**Visual data-association methods**    Data association has been explored in the context of visual object tracking (Luo et al., 2014; Xiang et al., 2015; Brasó & Leal-Taixé, 2020; Ma et al., 2019; Sun et al., 2019; Zhang et al., 2022). In these problems, there is typically a fixed visual field populated with many smoothly moving objects. This is an important special case of the general data-association. It enables some specialized techniques that take advantage of the fact that the observations of each object are typically smoothly varying in space-time, and incorporate additional visual appearance cues. In contrast, we are interested in exploring the general data-association problem where observations are not necessarily temporally correlated.

**Learning for data association**    There is relatively little work in the area of generalized data association, but Liu et al. (2019) provide a recent application of LSTMs to a rich version of the data association problem, in which batches of observations arrive simultaneously, with a constraint that each observation can be assigned to at most one object hypothesis. The sequential structure of the LSTM is used here not for recursive filtering, but to handle the variable numbers of observations and hypotheses. It is assumed that Euclidean distance is an appropriate metric and that the observation and state spaces are the same. Milan et al. (2017) combine a similar use of LSTM for data association with a recurrent network that learns to track multiple targets. It learns a dynamics model for the targets, including birth and death processes, but operates in simple state and observation spaces.

**Slot Based and Object Centric Learning**    Our approach to the dynamic entity monitoring task relies on the use of attention over a set of object hypothesis slots. Generic architectures for processing such slots can be found in (Vinyals et al., 2015; Lee et al., 2018), where we use (Lee et al., 2018) as a point of comparison for DAF-Net. We note that these architectures provide generic mechanisms to process sets of inputs, and lack the explicit structure from DAF we build into our model. Our individual hypothesis slots correspond to beliefs over object hypotheses, and thus also relates to existing work in object-centric scene learning. Such work has explored the discovery of factorized objects from both static scenes (Burgess et al., 2019; Greff et al., 2019; Locatello et al., 2020; Du et al., 2021; Kipf et al., 2022), but does not focus filtering and updating existing object hypotheses.

**Algorithmic priors for neural networks**    One final comparison is to other methods that integrate algorithmic structure with end-to-end neural network training. This approach has been applied to sequential decision making by Tamar et al. (2016), particle filters by Jonschkowski et al. (2018), and Kalman filters by Krishnan et al. (2015), as well as to a complex multi-module robot control system by Karkus et al. (2019). The results generally are much more robust than completely hand-built models and much more sample-efficient than completely unstructured deep-learning. We view our work as an instance of this general approach.

## 3    Problem formulation

The problem of learning to perform online data association requires careful formulation. When the DAF is executed online, it will receive a stream of input detections $z_1, \ldots z_T$ where $z_t \in \mathbb{R}^{d_z}$, and after each input $z_t$, it will output two vectors, $y_t = [y_{tk}]_{k \in (1..K)}$ and $c_t = [c_{tk}]_{k \in (1..K)}$, where $y_{tk} \in \mathbb{R}^{d_y}$, $c_{tk} \in (0, 1)$ and $\sum_k c_{tk} = 1$. The $y$ values in the output represent the predicted properties of the hypothesized objects and the $c$ values represent a measure of confidence in the hypotheses, in terms of the proportion of data that each one has accounted for. The maximum number of hypothesis "slots" is limited in advance to $K$. In some applications, the $z$ and $y$ values will be in the same space with the same representation, but this is not necessary.

We have training data representing $N$ different data-association problems, $\mathcal{D} = \{(z_t^{(i)}, m_t^{(i)})_{t \in (1..L_i)}\}_{i \in (1..N)}$, where each training example is an input/output sequence of length $L_i$, each element of which consists of a pair of input $z$ and $m = \{m_j\}_{j \in (1..J_t^{(i)})}$ which is a set of nominal object hypotheses representing the true

*current state* of objects that have actually been observed so far in the sequence. It will always be true that $m_t^{(i)} \subseteq m_{t+1}^{(i)}$ and $J_t^{(i)} \le K$.

Our objective is to train a recurrent computational model to perform DAF effectively in problems that are drawn from the same distribution as those in the training set. To do so, we formulate a model (described in section 4) with parameters $\theta$, which transduces the input sequence $z_1, \ldots, z_L$ into an output sequence $(y_1, c_1), \ldots, (y_L, c_L)$, and train it to minimize the following loss function:

$$\mathcal{L}(\theta; \mathcal{D}) = \sum_{i=1}^{N} \sum_{t=1}^{L_i} \mathcal{L}_{\text{obj}}(y_t^{(i)}, m_t^{(i)}) + \mathcal{L}_{\text{slot}}(y_t^{(i)}, c_t^{(i)}, m_t^{(i)}) + \mathcal{L}_{\text{sparse}}(c_t^{(i)}) \ .$$

The $\mathcal{L}_{\text{obj}}$ term is a *chamfer loss* (Barrow et al., 1977), which looks for the predicted $y$ that is closest to each actual $m_k$ and sums their distances, making sure the model has found a good, high-confidence representation for each true object:

$$\mathcal{L}_{\text{obj}}(y, m) = \sum_j \min_k \frac{1}{c_k + \epsilon} \|y_k - m_j\| \ .$$

The $\mathcal{L}_{\text{slot}}$ term is similar, but makes sure that each object the model has found is a true object, where we multiply by $c_k$ to not penalize for predicted objects in which we have low confidence:

$$\mathcal{L}_{\text{slot}}(y, c, m) = \sum_k \min_j c_k \|y_k - m_j\| \ .$$

The sparsity loss discourages the model from using multiple outputs to represent the same true object:

$$\mathcal{L}_{\text{sparse}}(c) = -\log\|c\| \ ,$$

and we theoretically show in Section 5 how this induces sparsity in confidences.

## 4   DAF-Nets

Inspired by the the basic form of classic DAF algorithms and the ability of modern neural-network techniques to learn complex models, we have designed the DAF-Net architecture for learning DAFs and a customized procedure for training it from data, inspired by several design considerations. First, because object hypotheses must be available after each individual input and because observations will generally be too large and the problem too difficult to solve from scratch each time, the network will have the structure of a recursive filter, with new memory values computed on each observation and then fed back for the next. Second, because the loss function is *set based*, that is, it doesn't matter what order the object hypotheses are delivered in, our memory structure should also be permutation invariant, and so the memory processing is in the style of an attention mechanism. Finally, because in some applications the observations $z$ may be in a representation not well suited for hypotheses representation and aggregation, the memory operates on a latent representation that is related to observations and hypotheses via encoder and decoder modules.

Figure 1 shows the architecture of the DAF-Net model and an illustration of its similarity to existing DAF approaches. The memory of the system is stored in $s$, which consists of $K$ elements, the $K$ hypotheses in DAF, each in $\mathbb{R}^{d_s}$; the length-$K$ vector $n$ of positive values encodes how many observations have been assigned to each slot during the execution so far. New observations are combined with the memory state, and the state is updated to reflect the passage of time by a neural network constructed from seven modules with trainable weights.

When an observation $z$ arrives, it is immediately **encode**d into a vector $e$ in $\mathbb{R}^{d_s}$, which is fed into subsequent modules. First, **attention** weights $w$ are computed for each hypothesis slot, using the encoded input and the existing content of that slot, representing the degree to which the current input "matches" the current value of each hypothesis in memory, mirroring the hypothesis matching procedure in DAFs. Since an observation typically matches only a limited number of hypotheses in DAFs, we force the network to commit to a sparse assignment of observations to object hypotheses while retaining the ability to effectively train with gradient

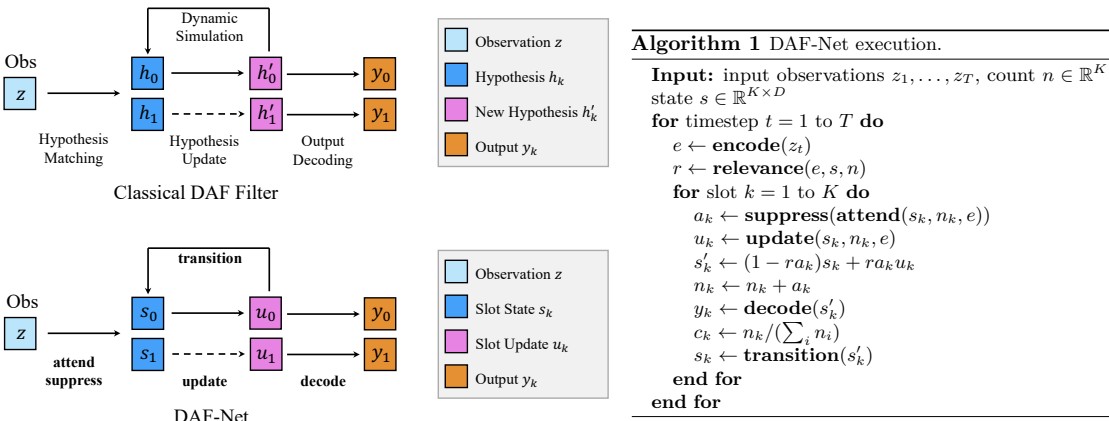

Figure 1: **Architecture and pseudocode of DAF-Net.** DAF-Net serves as a learned analogue of a DAF filter. The traditional hypothesis representation $h_k$ in a DAF-filter is replaced with a latent representation $s_k$. Hypothesis matching is replaced by sparse attention operators **suppress** and **attend**. Hypothesis updating is replaced by a **update** operator and dynamics simulation is replaced by a learned **transition** operators. Output decoding is replaced by a learned **decode** operator.

descent, the **suppress** module sets all but the top $M$ values in $w$ to 0 and renormalizes, to obtain the vector $a$ of $M$ values that sum to 1:

$$w_k = \frac{\exp(\textbf{attend}(s_k, n_k, e))}{\sum_{j=0}^{n} \exp(\textbf{attend}(s_j, n_k, e))} \quad ; \quad a = \textbf{suppress}(w) \quad .$$

The $a$ vectors are integrated to obtain $n$, which is normalized to obtain the output confidence $c$.

Mirroring hypothesis updates in DAFs, the **update** module also operates on the encoded input and the contents of each hypothesis slot, producing a hypothetical update of the hypothesis in that slot under the assumption that the current $z$ is an observation of the object represented by that slot; so for all slots $k$,

$$u_k = \textbf{update}(s_k, n_k, e) \quad .$$

To enable the rejection of outlier observations, a scalar **relevance** value, $r \in (0, 1)$, is computed from $s$ and $e$; this value modulates the degree to which slot values are updated, and gives the machine the ability to ignore or downweight an input. It is computed as

$$r = \textbf{relevance}(e, s, n) = \text{NN}_2(\underset{k=1}{\overset{K}{\text{avg}}} \text{NN}_1(e, s_k, n_k)) \quad ,$$

where $\text{NN}_1$ is a fully connected network with the same input and output dimensions and $\text{NN}_2$ is a fully connected network with a single sigmoid output unit. The attention output $a$ and relevance $r$ are now used to decide how to combine all possible slot-updates $u$ with the old slot values $s_t$ using the following fixed formula for each slot $k$:

$$s'_{tk} = (1 - ra_k)s_{tk} + ra_k u_k \quad .$$

Because most of the $a_k$ values have been set to 0, this results in a sparse update which will ideally concentrate on a single slot to which this observation is being "assigned".

To obtain outputs, slot values $s'_t$ are then **decoded** into the outputs, $y$, using a fully connected network:

$$y_k = \textbf{decode}(s'_{tk}) \quad .$$

Finally, to simulate transition updates in DAFs and to handle the setting in which object state evolves over time, we add a **transition** module, which computes the state $s_{t+1}$ from the new slot values $s'_t$ using an additional neural network:

$$s_{t+1_k} = \textbf{transition}(s'_t)_k \quad .$$

These values are then fed back, recurrently, as inputs to the overall system.

Given a data set $\mathcal{D}$, we train the DAF-Net model end-to-end to minimize loss function $\mathcal{L}$, with a slight modification. We find that including the $\mathcal{L}_{\text{sparse}}$ term from the beginning of training results in poor learning, but adopting a training scheme in which the $\mathcal{L}_{\text{sparse}}$ is first omitted then reintroduced over training epochs, results in reliable training that is efficient in both time and data.

## 5 Theoretical Analysis

In this section, we study the extent to which DAF-Net may learn to construct an optimal DAF. First, we illustrate how our underlying slot-based architectures enables more efficient learning under the framework of algorithmic alignment introduced by Xu et al. (2019). We further illustrate how $\mathcal{L}_{\text{sparse}}$ induces sparsity across slot values.

First we analyze the underlying sample complexity of learning DAF-Net, which represents a set of belief states as a set of slots, compared to a network which explicitly represents intermediate beliefs as a single flattened vector. We use the notion of sample complexity analysis introduced by Xu et al. (2019). For simplicity, we consider the online data-association problem of clustering, where each network takes as input an observation $z$ and a set of $K$ previously predicted cluster centers $y_k$, and must correctly predict the updated state of each observed cluster center. In addition, we assume for purposes of applying existing theoretical results that the system is supervised with the ground-truth clusters at each step.

**Proposition 1.** *Consider the problem of performing one step of an online clustering algorithm, in which $K$ current cluster centers, $y_1, \ldots y_K$ and a new element $x$ are the inputs and the updated cluster centers $y'_1, \ldots y'_K$ are the outputs. We consider two different architectures: (1) a generic MLP, which we denote as $f(x)$, in which the inputs $y_1, \ldots, y_K, x$ are simply concatenated into an input vector, and the outputs $y'_1, \ldots y'_K$ are an output vector and (2) an instance of DAF-Net with no transition module which we denote as $g(x)$. The sample complexity of learning the MLP for $K$ clusters is $K$ times the sample complexity for learning the DAF-Net.*

*Proof.* The sample complexity of learning to approximate a function $h : \mathbb{R}^d \to \mathbb{R}^m$, where $h^{(i)}(x) = \sum_j \alpha_j^{(i)} (\beta_j^{(i)} x)^{p_j^{(i)}}$ with MLP to error less than $\epsilon$ with probability $1 - \delta$ is asymptotically given by (Xu et al., 2019):

$$\mathcal{C}(h, \epsilon, \delta) = O\left( \frac{\max_i \sum_{j=1}^{K} p_j^{(i)} \|\alpha_j^{(i)}\| \|\beta_j^{(i)}\|_2^{p_j^{(i)}} + \log(m/\delta)}{(\epsilon/m)^2} \right). \tag{1}$$

The above expression implies that the sample complexity of learning a neural network to approximate a function $h(x)$ is proportional to the underlying number of polynomial terms needed to represent $h(x)$.

A sketch of our proof is that the number of polynomial terms necessary to accurately represent the clustering procedure, using a MLP $f(x)$, is $K$ times more than the number of terms using DAF-Net $g(x)$. This statement is true because $g(x)$ only needs to learn the clustering operation *per slot* as the underlying computation is replicated across slots, while $f(x)$ needs to learn the clustering operation for all slots simultaneously.

Concretely, consider learning a simplified cluster-update function, $h(z, y_k) = (1 - w_k) * y_k + w_k * z$, for each cluster center $k$, where $w_k$ is a constant $w_k = \frac{1/\|z - y_k\|}{\sum_k 1/\|z - y_k\|}$ predicted by a fixed network, determining the extent to which $z$ should be assigned to $y_k$. The DAF-Net architecture, operates independently on each cluster, and is required to approximate a function $h$ consisting of two polynomial terms. However, the MLP architecture operates jointly on all of the clusters, without any parameter tying, and would require $2K$ polynomial terms, as it needs to jointly approximate the function $h$ across all cluster centers. Thus it is more sample efficient to learn DAF-Net. $\square$

Our previous result illustrates the benefits of learning filters using the architecture of DAF-Net. Next, we illustrate that our proposed sparsity loss accurately induces sparsity in the assignment of each object to potential clusters.

**Proposition 2.** *The sparsity loss*

$$\mathcal{L}_{\text{sparse}}(\boldsymbol{c}) = -\log\|\boldsymbol{c}\| \ , \tag{2}$$

*where $\sum_i \boldsymbol{c}_i = 1$, $\boldsymbol{c}_i \geq 0$ is minimized when a particular element $\boldsymbol{c}_k = 1$ and maximized when individual confidences are equal.*

*Proof.* Recall that $\|\mathbf{c}\|$ is a convex function. The confidence vector $c$ defines a simplex, with each element being non-negative, and whose elements sum up to one. The maximum of a convex function over a simplex must occur at the vertices. Each vertex in this simplex is symmetric, and corresponds to a value of $\boldsymbol{c}$ with hypothesis $k$ having confidence $\boldsymbol{c}_k = 1$ and other confidences corresponding to 0. In contrast, the minimum corresponds to a stationary point at the Lagrangian of the loss. The Lagrangian of the loss $L(c, \lambda)$ is

$$L(c, \lambda) = \sum_i c_i^2 + \lambda(\sum_i c_i - 1). \tag{3}$$

By taking the gradient of the above expression, we find that the stationary value corresponds to each $c_i$ being equal. Since the function is convex, this corresponds to the minimum of $\|c\|$. Thus $\mathcal{L}_{\text{sparse}}(\mathbf{c})$ is maximized when individual confidences are equal and minimized when individual confidences are sparse. □

## 6 Empirical Results

We evaluate DAF-Net on several different *data association* tasks. First, we consider a simple online clustering task and validate the underlying machinery of DAF-Net as well as its ability to generalize at inference time to differences in (a) the number of actual objects, (b) the number of hypothesis slots and (c) the number of observations. Next, we evaluate the performance of DAF-Net on dynamic domains. Finally, we evaluate the performance of DAF-Net on an image domain in which the underlying observation space is substantially different from the hypothesis space.

- DAF-Net outperforms non-learning clustering methods, even those that operate in batch mode rather than online, because those methods cannot learn from experience to take advantage of information about the distribution of observations and true object properties (Tables 1, 1 and 6).
- DAF-Net outperforms clustering methods that can learn from previous example problems when data is limited, because it provides useful structural bias for learning (Table 1, 1 and 6).
- DAF-Net generalizes to differences between training and testing in (a) the numbers of actual objects (Figure 6 and 3), (b) the numbers of hypothesis slots (Table 2) and (c) the number of observations (Figure 3).
- DAF-Net works when significant encoding and decoding are required (Table 6).
- DAF-Net is able to learn dynamics models and observation functions for the setting when the entities are moving over time (Table 5), nearly matching the performance of strong data association filters with known ground-truth models.

**Baselines and Metrics** In each domain, we compare DAF-Net to online learned baselines of LSTM (Hochreiter & Schmidhuber, 1997) and Set Transformer (Lee et al., 2018) (details in A.3), as well as to task-specific baselines. All learned network architectures are structured to use $\sim 50000$ parameters. Unless otherwise noted, models except DAF-Net are given and asked to predict the ground truth number of components $K$, while DAF-Net uses 10 hypothesis slots. Results are reported in terms of MSE error $\frac{1}{K}\min_j\|y_k - m_j\|$ (with respect to the most confident $K$ hypotheses for DAF-Net).

### 6.1 Online Clustering

**Setup.** To check the basic operation of the model and understand the types of problems for which it performs well, we tested in simple clustering problems with the same input and output spaces, but different types of data distributions, each a mixture of three components. We train on 1000 problems with observation sequences of length 30 drawn from each problem distribution and test on 5000 from the same distribution. In every case, the means of the three components are drawn at random for each problem. We provide precise details about distributions in Section A.1.

| Model | Online | Learned | Normal | Elongated | Mixed | Angular | Noise |
|---|---|---|---|---|---|---|---|
| DAF-Net | + | + | **0.157** | **0.191** | **0.184** | **0.794** | **0.343** |
| Set Transformer | + | + | 0.407 | 0.395 | 0.384 | 0.794 | 0.424 |
| LSTM | + | + | 0.256 | 0.272 | 0.274 | 0.799 | 0.408 |
| VQ | + | - | 0.173 | 0.195 | 0.191 | 0.992 | 0.947 |
| Set Transformer | - | + | 0.226 | 0.248 | 0.274 | **0.816** | **0.406** |
| Slot Attention | - | - | 0.254 | 0.267 | 0.268 | 0.823 | 0.504 |
| K-means++ | - | - | **0.103** | **0.139** | **0.135** | 0.822 | 1.259 |
| GMM | - | - | 0.113 | 0.141 | 0.136 | 0.865 | 1.207 |

Table 1: **Quantitative Results on Online Clustering.** Comparison of performance on clustering performance across different distributions. Reported error is the L2 distance between predicted and ground truth means. Methods in the bottom half of table operate on observations in a single batch and thus are not directly comparable.

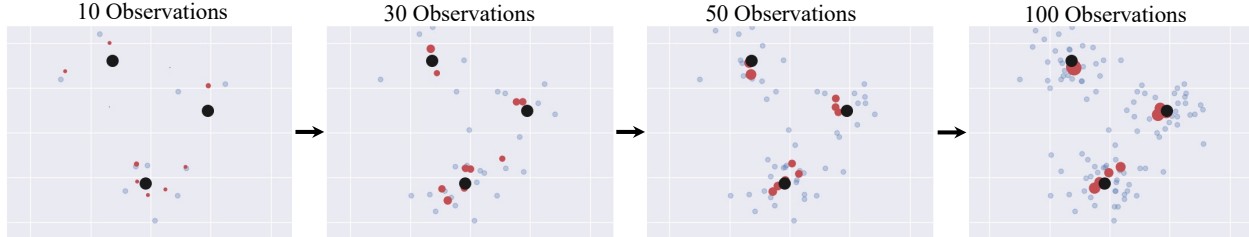

Figure 2: **Qualitative Visualization of DAF-Net.** Illustration of DAF-Net execution on the *Normal* distribution setting. Decoded value of hypothesis (with size corresponding to confidence) shown in red, with ground truth clusters in black. Observations are shown in blue.

1. *Normal*: Each component is a 2D Gaussian with fixed identical variance across each individual dimension and across distributions. This is a basic "sanity check."
2. *Elongated*: Each component is a 2D Gaussian, where the variance along each dimension is drawn from a uniform distribution, but fixed across distributions.
3. *Mixed*: Each component is a 2D Gaussian, with fixed identical variance across each individual dimension, but with the variance of each distribution drawn from a uniform distribution.
4. *Angular*: Each component is a 2D Gaussian with identical variance across dimension and distribution, but points above $\pi$ are wrapped around to $-\pi$ and points below $-\pi$ wrapped to $\pi$
5. *Noise*: Each component has 2 dimensions parameterized by Gaussian distributions, but with the values of the remaining 30 dimensions drawn from a uniform centered at 0.

**Results.** We compare our approach to each of the baselines for the five problem distributions in Table 1. The results in this table show that on *Normal*, *Mixed*, and *Elongated* tasks, DAF-Net performs better than

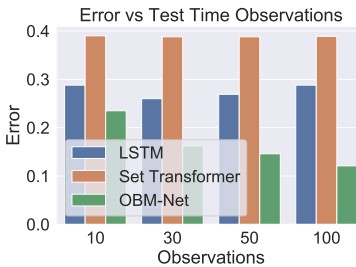

Figure 3: **Generalization with Increased Observations.** Plot of LSTM, Set Transformer and DAF-Net errors when executed *at test time* on different number of observations from the *Normal* distribution. With increased observations, DAF-Net error continues to decrease while other approaches obtain higher error.

| Model | Slots | Ground Truth Clusters | | |
|---|---|---|---|---|
| | | 3 | 5 | 7 |
| DAF-Net | 10 | **0.162** | 0.214 | 0.242 |
| | 20 | 0.175 | **0.195** | 0.213 |
| | 30 | 0.188 | 0.197 | **0.205** |
| Set Transformer | - | 0.261 | 0.279 | 0.282 |
| Vector Quantization | - | 0.171 | 0.199 | **0.205** |

Table 2: **Generalization with Different Hypothesis Slots.** Error of DAF-Net, when executed *at test time* with a different number of hypothesis slots on test distributions with different numbers of ground true components. In all cases, DAF-Net is trained on 3-component problems with 10 slots. DAF-Net achieves good performance with novel number of hypothesis slots, and outperforms different instances of the Set Transformer trained with the ground truth number of cluster components as well as vector quantization.

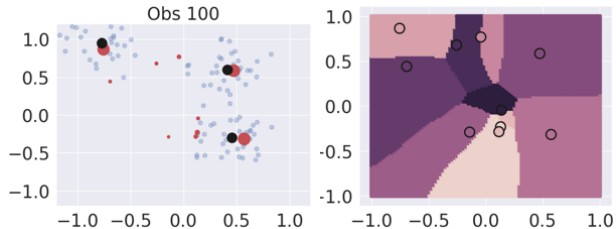 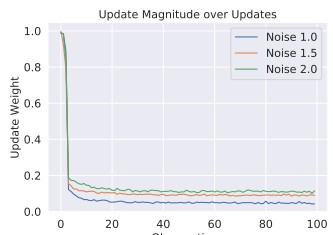

Figure 4: **Visualization of Attention Weights.** Plot of decoded values of slots (in red) with confidence shown by the size of dot (left), and what slot each input assigns the highest attention towards (right) (each slot is colored differently, with assigned inputs colored in the same way). Note alignment of regions on the right with the decoded value of a slot on the left.

Figure 5: **Visualization of Relevance Weights.** Plots of the magnitude of relevance weights with increased observation number on different distributions with differing standard deviation (noise).

learned and constructed online clustering algorithms, but does slightly worse than offline clustering algorithms. Such discrepancy in performance is to be expected due to the fact that DAF-Net is running and being evaluated online. On the *Angular* and *Noise* tasks, DAF-Net is able to learn a useful metric for clustering and outperforms both offline and online alternatives.

Next, we provide a qualitative illustration of execution of DAF-Net on the *Normal* clustering task in Figure 2 as a trajectory of observations are seen. We plot the decoded values of hypothesis slots in red, with size scaled according to confidence, and ground-truth cluster locations in black. DAF-Net is able to selectively refine slot clusters to be close to ground truth cluster locations even with much longer observation sequences than it was trained on.

**Baselines.** In addition to baselines discussed earlier, we further compare with clustering specific baselines of **Batch, non-learning**: K-means++ (Arthur & Vassilvitskii, 2007) and expectation maximization (EM) (Dempster et al., 1977) on a Gaussian mixture model (SciKit Learn implementation); **Online, non-learning**: vector quantization (Gray, 1984). We further provide a comparison to a recent concurrent work (Locatello et al., 2020) which also utilizes attention to obtains slots of objects.

**Generalization.** We next assess the ability of DAF-Net to generalize at inference time to differences in the number of input observations as well as differences in the underlying number of hypothesis slots used on the *Normal* distribution. In Figure 3, we plot the error of LSTM, Set Transformer, and DAF-Net as a function of the number observations seen at inference time. We find that when DAF-Net is given more observations then seen during training time (all models are trained with observations of length 30), it is able to further improve its performance, while both LSTM and Set Transformer results suffer. We believe that such generalization ability is due to the inductive bias added to DAF-Net. We provide additional analysis of this generalization across all distributions in Table 7 and find similar results.

In Table 2, we investigate the ability of DAF-Net to generalize at *inference time* to increases in both the number of hypothesis slots and the underlying number of mixture components from which observations are drawn. We compare to the Set Transformer and to VQ, both of which *know the correct number of components at inference time.* We find that DAF-Net generalizes well to increases in hypothesis slots, and exhibits improved performance with large number of underlying components, performing comparably to or better than the VQ algorithm. We further note that none of the *learning* baselines can adapt to different numbers cluster components at inference time, but find that DAF-Net outperforms the Set Transformer even when it is trained on the ground truth number of clusters in the test.

**Submodule Visualization.** We find that individual modules learned by DAF-Net are interpretable. We visualize the attention weights of each hypothesis slot in Figure 4 and find that each hypothesis slot learns to attend to a local region next to the value it decodes to. We further visualize a plot of relevance weights in Figure 5 across an increasing number of observations where individual observations are drawn from distributions with different levels of noise with respect to cluster centers. We find that as DAF-Net sees more observations, the relevance weight of new observations decreases over time, indicating that DAF-Net

| Sparsity | Learned Memory | Supression | Relevance | Observations | | | |
|---|---|---|---|---|---|---|---|
| | | | | 10 | 30 | 50 | 100 |
| − | − | − | − | 0.382 | 0.452 | 0.474 | 0.487 |
| + | − | − | − | 0.384 | 0.412 | 0.423 | 0.430 |
| + | + | − | − | 0.335 | 0.357 | 0.366 | 0.387 |
| + | + | + | − | 0.279 | 0.274 | 0.278 | 0.282 |
| + | + | + | + | **0.238** | **0.157** | **0.137** | **0.131** |

Table 3: **Abalation Analysis.** We ablate each component of DAF-Net on the *Normal* distribution . When learned memory is ablated, DAF-Net updates states based on observed values (appropriate in the *Normal* distribution dataset).

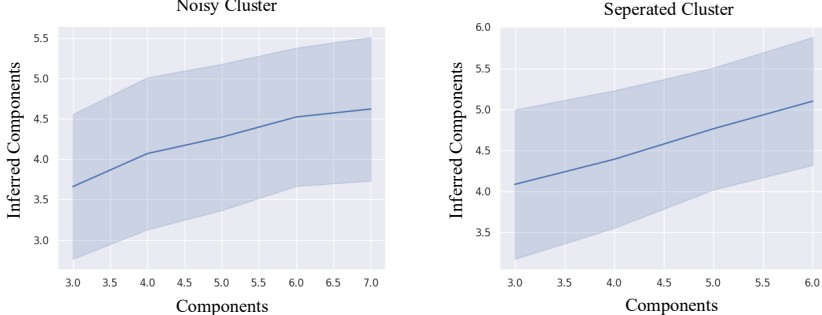

Figure 6: **Generalization to Increased Cluster Number.** Plots of inferred number of components using a confidence threshold in DAF-Net compared to the ground truth number of clusters (DAF-Net is trained on only 3 clusters). We consider two scenarios, a noisy scenario where cluster centers are randomly drawn from -1 to 1 (left) and a scenario where all added cluster components are well seperated from each other (right). DAF-Net is able to infer more clusters in both scenarios, with better performance when cluster centers are more distinct from each other.

learns to pay the most attention towards the first set of observations it sees. In addition, we find that in distributions with higher variance, the relevance weight decreases more slowly, as later observations are now more informative in determining cluster centers.

**Ablation.** We ablate each component of DAF-Net and present results in Table 3 on the *Normal* distribution. We test removing $\mathcal{L}_{\text{sparse}}$ (sparsity), removing learned slot embeddings (learned memory) — where instead, in individual hypothesis slots, we store the explicit values of inputs, removing the **suppress** modules (suppression) and removing the **relevance** module (relevance). We find that each of our proposed components enables better performance on the underlying clustering task. Interestingly, we further find that the addition of **relevance** enables our approach to generalize at test time to larger numbers of observations.

**Inferring Object Number.** In contrast to other algorithms, DAF-Net learns to predict both a set of object properties $y_k$ of objects and a set of confidences $c_k$ for each object. This corresponds to the task of both predicting the number of objects in a set of observations, as well as the associated object properties. We evaluate the ability of DAF-Net to regress object number at *test time* in scenarios where the number of objects (underlying clusters) is different than that of training. We evaluate on the *Normal* distribution with a variable number of component distributions, and measure inferred components through a threshold confidence. DAF-Net is trained on a dataset with 3 underlying components. We find in Figure 6 that DAF-Net is able to infer the presence of more component distributions (as they vary from 3 to 6), with improved performance when cluster centers are sharply separated (right figure of Figure 6).

**Performance on More Clusters.** We find DAF-Net also exhibits good performance when trained and tested on domains with a larger number of slots/clusters. To test this, we utilize the *Normal* distribution setting, but generate underlying training input observations from a total of 30 difference components, and train DAF-Net with a total of 30 slots. We train DAF-Net with 50 observations, and measure performance at inferring cluster centers with between 50 to 100 observations. We report performance in Table 4 and find that DAF-Net obtains good performance in this setting, out-performing the strong online baseline VQ, and performing similarly to K-means++ which directly operates on all input observations at once.

| Model | Online | Observations | | | |
|---|---|---|---|---|---|
| | | 50 | 65 | 80 | 100 |
| DAF-Net | + | **0.158** | **0.154** | **0.151** | **0.147** |
| VQ | + | 0.162 | 0.157 | 0.153 | 0.148 |
| K-means++ | - | **0.155** | **0.151** | **0.148** | **0.146** |
| GMM | - | 0.156 | 0.151 | 0.149 | 0.147 |

Table 4: **Performance on Large Number of Clusters.** Comparison of performance on *Normal* distribution, when underlying distributions have a large number of components. We use 30 components, and train models with 50 observations. Each cluster observation and center is drawn between -1 and 1, with reported error as the L2 distance between predicted and ground truth means.

| Model | Observations | | | |
|---|---|---|---|---|
| | 10 | 20 | 30 | 40 |
| DAF-Net | **0.415** | 0.395 | 0.382 | 0.394 |
| Set Transformer | 0.699 | 0.701 | 0.854 | 1.007 |
| LSTM | 0.422 | 0.400 | 0.445 | 0.464 |
| JPDA (ground truth) | 0.683 | **0.372** | **0.362** | **0.322** |

Table 5: **Performance on Dynamic Objects.** Comparison of different methods on estimating the state of 3 dynamically moving objects. All learning models are trained with 1000 sequences of 30 observations. We report MSE error. JPDA uses the ground-truth observation and dynamics models. JPDA is outperformed by learned approaches with 10 observations, as these approaches are able to average over possible outputs to minimize MSE error.

## 6.2 Dynamic Domains

Now we study the ability of DAF-Net to perform data association in a dynamic setting and compare its performance with that of a classical data-association filter.

**Setup.** We evaluate performance of data association monitoring using moving 2D objects. A *problem* involves a trajectory of observations $z$ of the $K$ dynamically moving objects, with desired $y$ values being the underlying object positions. Objects evolve under a linear Gaussian dynamics model, with a noisy observation of a single object observed at each step (details in Appendix A.1). This task is typical of tracking problems considered by DAF. All learning-based models are trained on observation sequences of length 30. To perform well in this task, a model must discover that it needs to estimate the latent velocity of each object, as well as learn the underlying dynamics and observation models. We utilize $K = 3$ for our experiments.

**Baselines.** We compare with the de-facto standard method, Joint Probabilistic Data Association (JPDA) (Bar-Shalom et al., 2009), which uses hand-built ground-truth models (serving as an oracle). We further compare with our learned online baselines of Set Transformer (Lee et al., 2018) and LSTM (Hochreiter & Schmidhuber, 1997) methods.

**Result.** Quantitative performance, measured in terms of prediction error on true object locations, is reported in Table 5. We can see that the Set Transformer cannot learn a reasonable model at all. The LSTM performs reasonably well for short (length 30) sequences but quickly degrades relative to DAF-Net and JPDA as sequence length increases. We note that DAF-Net performs comparably to, but just slightly worse than, JPDA. *This is strong performance because DAF-Net is generic and can be adapted to new domains given training data without the need to hand-design the models in JPDA.* We believe that these gains are due to the inductive biases built into our architecture.

## 6.3 Image-based domains

Finally, we validate the ability of DAF-Net to perform data association on image inputs, which requires DAF-Net to synthesize a latent representation for slots, and learn to perform association, update, and transition operations in that space.

**Setup.** We experiment with two separate image-based domains, each consisting of a set of similar entities (2D digits or 3D airplanes). We construct data association *problems* by selecting $K$ objects in each domain,

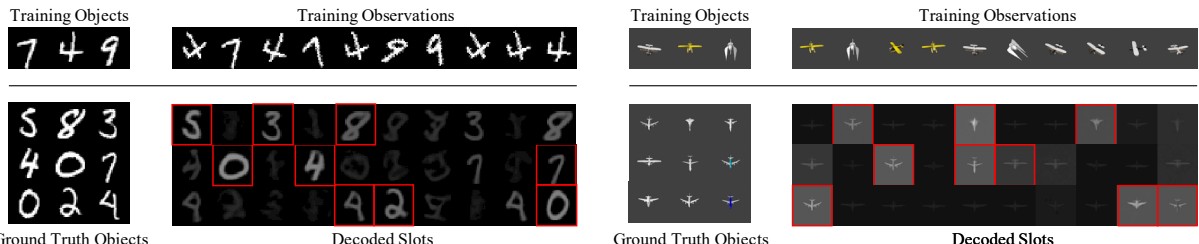

Figure 7: **Qualitative Visualization of DAF-Net Execution on Images.** Qualitative visualization of two image-based association tasks (left: MNIST, right: airplanes). At the top of each is an example training problem, illustrated by the true objects and an observation sequence. Each of the next rows shows an example test problem, with the ground truth objects and decoded slot values. The three highest-confidence hypotheses for each problem are highlighted in red, and correspond to ground-truth objects.

| Model | Learned | MNIST | | | | Airplanes | | | |
|---|---|---|---|---|---|---|---|---|---|
| Observations | | 10 | 30 | 50 | 100 | 10 | 30 | 50 | 100 |
| DAF-Net | + | **7.143** | **5.593** | **5.504** | **5.580** | **4.558** | **4.337** | **4.331** | **4.325** |
| LSTM | + | 9.980 | 9.208 | 9.166 | 9.267 | 5.106 | 4.992 | 4.983 | 4.998 |
| K-means | + | 13.596 | 12.505 | 12.261 | 12.021 | 7.246 | 6.943 | 6.878 | 6.815 |

Table 6: **Quantitative Results on Image Domain.** Comparison of entity-monitoring performance on MNIST and Airplane datasets across 10, 30, 50, 100 observations. For DAF-Net, LSTM and K-means we use a convolutional encoder/decoder trained on the data. We train models with 30 observations and report MSE error.

with the desired $y$ values being images of those objects in a canonical viewpoint. An input observation sequence is generated by randomly selecting one of those $K$ objects, and generating an observation $z$ corresponding to a random viewpoint of the object. Our two domains are: (1) **MNIST**: Each object is a random image in MNIST, with observations corresponding to rotated images, and the desired outputs being the un-rotated images; (2) **Airplane**: Each object is a random object from the Airplane class in ShapeNet (Chang et al., 2015), with observations corresponding to airplane renderings (using Blender) at different viewpoints and the desired outputs the objects rendered in a canonical viewpoint. For MNIST, we use the training set to construct the training problems, and the images in the non-overlaping test set to construct the test problems. For the Airplane dataset, we use 1895 airplanes to construct the training problems, and 211 different airplanes to construct the test problems. Each airplane is rendered with 300 viewpoints.

**Baselines.** In addition to our learned baselines, we compare with a task specific baseline, batch K-means, in a latent space that is learned by training an autoencoder on the observations. In this setting, we were unable to train the Set Transformer stably and do not report results for it.

**Results.** In Table 6, we find that our approach significantly outperforms other comparable baselines in both accuracy and generalization. We further visualize qualitative predictions from our model in Figure 7. We find that our highest confidence decoded slots correspond to ground truth objects.

# 7 Conclusion

This work has demonstrated that using algorithmic bias inspired by a classical solution to the problem of filtering to estimate the state of multiple objects simultaneously, coupled with modern machine-learning techniques, we can arrive at solutions that learn to perform and generalize well from a comparatively small amount of training data.

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

We further provide experimental and architecture details of DAF-Net in Section A. We further provide full clustering results in Table 7.

## A  Experimental Details

In this section, we provide details of our experimental approach. We first discuss the details of datasets used in Section A.1. Next, we provide the model architectures used in Section A.2. Finally, we provide details on the baselines we compare with in Section A.3.

### A.1  Dataset Details

We first provide detailed experimental settings for each of the datasets considered in the paper.

**Online Clustering.**    In online clustering, we utilize observations drawn from the following distributions, where Gaussian centers are drawn uniformly from -1 to 1.

1. *Normal*: Each 2D Gaussian has standard deviation 0.2. The normal setting is illustrated in Figure **??**.
2. *Mixed*: Each distribution is a 2D Gaussian, with fixed identical variance across each individual dimension, but with the standard deviation of each distribution drawn from a uniform distribution from (0.04, 0.4).
3. *Elongated*: Each distribution is a 2D Gaussian, where the standard deviation along each dimension is drawn from a uniform distribution from (0.04, 0.4), but fixed across distributions.
4. *Angular*: Each distribution is a 2D Gaussian with identical standard deviation across dimension and distribution, but points above $\pi$ are wrapped around to $-\pi$ and points below $-\pi$ wrapped to $\pi$. Gaussian means are selected between $(-\pi, -2\pi/3)$ and between $(2\pi/3, \pi)$. The standard deviation of distributions is $0.3 * \pi$.
5. *Noise*: Each distribution has 2 dimensions parameterized by Gaussian distributions with standard deviation 0.5, but with the values of the remaining 30 dimensions drawn from a uniform distribution from $(-1, 1)$.

**Dynamic Domains.**    Next, in the dynamics domain, we implement our dataset using the StoneSoup library*. We initialize the location of each cluster from a Gaussian distribution with standard deviation 1.5 and initialize velocity in each directory from a Gaussian distribution with standard deviation of 0.02. At each timestep, Gaussian noise is added to velocities with magnitude 1e-4. Our JPDA implementation is also from the StoneSoup library.

**Image Domains.**    In the image domain, for MNIST, we use the 50000 images in the training set to construct the training problems, and the 10000 images in the non-overlapping test set to construct the test problems. For the Airplane dataset, we use 1895 airplanes to construct the training problems, and 211 different airplanes to construct the test problems. Each airplane is rendered with 300 viewpoints.

### A.2  Model/Baseline Architectures

We provide the overall architecture details for the LSTM in Figure 8a, for the Set Transformer in Figure 8b and DAF-Net in Figure 9a. For image experiments, we provide the architecture of the encoder in Figure 10a and decoder in Figure 10b. Both LSTM, DAF-Net, and autoencoding baselines use the same image encoder and decoder. For robotics experiments, we provide the architecture of the shape decoder in Figure **??**.

In DAF-Net memory, the function update($s_k, n_k, e$) is implemented by applying a 2 layer MLP with hidden units $h$ which concatenates the vectors $s_k, n_k, e$ as input and outputs a new state $u_k$ of dimension $h$. The value $n_k$ is encoded using the function $\frac{1}{1+n_k}$, to normalize the range of input to be between 0 and 1. The function attend($s_k, n_k, e$) is implemented in an analogous way to update, using a 2 layer MLP that outputs a single real value for each hypothesis slot.

For the function relevance($s_k, n_k, e$), we apply $\text{NN}_1$ per hypothesis slot, which is implemented as a 2 layer MLP with hidden units $h$ that outputs a intermediate state of dimension $h$. $(s_k, n_k, e)$ is fed into $\text{NN}_1$ in an

---

*https://stonesoup.readthedocs.io/en/v0.1b3/stonesoup.html

analogous manner to update. $NN_2$ is applied to average of the intermediate representations of each hypothesis slot and is also implemented as a 2 layer MLP with hidden unit size $h$, followed by a sigmoid activation. We use the ReLU activation for all MLPs. $NN_3$ is represented is GRU, which operates on the previous slot value.

### A.3 Baseline Details

All baseline models are trained using prediction slots equal to the ground truth of components. To train the Set Transformer to act in an online manner, we follow the approach in (Santoro et al., 2018) and we apply the Set Transformer sequentially on the concatenation of an input observation with the current set of hypothesis slots. Hypothesis slots are updated based off the new values of the slots after applying self-attention (Set Transformer Encoder). Hypothesis slots are updated based off the new values of the slots after applying self-attention (Set Transformer Encoder). We use the Chamfer loss to train baseline models, with confidence set to 1.

| Dense → h |
|---|
| Dense → h |
| LSTM(h) |
| Dense → h |
| Dense → output |

(a) The model architecture of the LSTM baseline. The hidden dimension $h$ used is 96 for synthetic Gaussian distributions and 128 for Image datasets. For image experiments, the first 2 and last 2 fully connected layers are replaced with image encoders and decoders.

| Dense → h |
|---|
| Dense → h |
| Set Transformer Encoder |
| Set Transformer Decoder |

(b) The model architecture of the Set Transformer baseline. The hidden dimension $h$ is 48 for the synthetic Gaussian distributions. We use the encoder and decoder defined in (Lee et al., 2018) with 4 heads and hidden dimension $h$.

Figure 8: Architecture of baseline models.

| Dense → h |
|---|
| Dense → h |
| DAF-Net Memory |
| Dense → h |
| Dense → output |

(a) The model architecture of DAF-Net. The hidden dimension $h$ is 64 is for synthetic Gaussian distributions and 128 for the image/robotics experiments. For image experiments, the first and last 2 linear layers are replaced with convolutional encoders and decoders.

Figure 9: Architecture of DAF-Net.

| 5x5 Conv2d, 32, stride 2, padding 2 |
|---|
| 3x3 Conv2d, 64, stride 2, padding 1 |
| 3x3 Conv2d, 64, stride 2, padding 1 |
| 3x3 Conv2d, 64, stride 2, padding 1 |
| 3x3 Conv2d, 128, stride 2, padding 1 |
| Flatten |
| Dense → h |

(a) The model architecture of the convolutional encoder for image experiments.

| Dense → 4096 |
|---|
| Reshape $(256, 4, 4)$ |
| 4x4 Conv2dTranspose, 128, stride 2, padding 1 |
| 4x4 Conv2dTranspose, 64, stride 2, padding 1 |
| 4x4 Conv2dTranspose, 64, stride 2, padding 1 |
| 4x4 Conv2dTranspose, 64, stride 2, padding 1 |
| 3x3 Conv2d, 3, stride 1, padding 1 |

(b) The model architecture of the convolutional decoder for image experiments.

Figure 10: The model architecture of the convolutional encoder and decoder for image experiments.

| Type | Model | Online | Observations | | | |
|---|---|---|---|---|---|---|
| | | | 10 | 30 | 50 | 100 |
| Normal | DAF-Net | + | 0.235 | 0.162 | 0.146 | 0.128 |
| | Set Transformer | + | 0.390 | 0.388 | 0.388 | 0.389 |
| | LSTM | + | 0.288 | 0.260 | 0.269 | 0.288 |
| | VQ | + | 0.246 | 0.172 | 0.147 | 0.122 |
| | Set Transformer | + | 0.295 | 0.261 | 0.253 | 0.247 |
| | K-means++ | - | 0.183 | 0.107 | 0.086 | 0.066 |
| | GMM | - | 0.189 | 0.118 | 0.087 | 0.067 |
| Mixed | DAF-Net | + | 0.255 | 0.184 | 0.164 | 0.147 |
| | LSTM | + | 0.306 | 0.274 | 0.284 | 0.290 |
| | Set Transformer | + | 0.415 | 0.405 | 0.407 | 0.408 |
| | VQ | + | 0.262 | 0.192 | 0.169 | 0.145 |
| | Set Transformer | - | 0.309 | 0.274 | 0.266 | 0.261 |
| | K-means++ | - | 0.206 | 0.135 | 0.105 | 0.088 |
| | GMM | - | 0.212 | 0.136 | 0.105 | 0.079 |
| Enlongated | DAF-Net | + | 0.258 | 0.192 | 0.173 | 0.161 |
| | LSTM | + | 0.314 | 0.274 | 0.288 | 0.300 |
| | Set Transformer | + | 0.394 | 0.391 | 0.394 | 0.394 |
| | VQ | + | 0.265 | 0.194 | 0.172 | 0.149 |
| | Set Transformer | - | 0.309 | 0.244 | 0.240 | 0.232 |
| | K-means++ | - | 0.213 | 0.139 | 0.113 | 0.092 |
| | GMM | - | 0.214 | 0.141 | 0.112 | 0.086 |
| Rotation | DAF-Net | + | 0.892 | 0.794 | 0.749 | 0.736 |
| | LSTM | + | 0.799 | 0.796 | 0.795 | 0.794 |
| | Set Transformer | + | 0.793 | 0.794 | 0.782 | 0.782 |
| | VQ | + | 0.956 | 1.000 | 1.000 | 0.984 |
| | Set Transformer | - | 0.815 | 0.784 | 0.779 | 0.772 |
| | K-means++ | - | 0.827 | 0.834 | 0.823 | 0.802 |
| | GMM | - | 0.842 | 0.875 | 0.867 | 0.848 |
| Noise | DAF-Net | + | 0.375 | 0.343 | 0.338 | 0.334 |
| | LSTM | + | 0.419 | 0.406 | 0.405 | 0.407 |
| | Set Transformer | + | 0.434 | 0.424 | 0.425 | 0.424 |
| | VQ | + | 1.479 | 0.948 | 0.826 | 0.720 |
| | Set Transformer | - | 0.436 | 0.407 | 0.398 | 0.394 |
| | K-means++ | - | 1.836 | 1.271 | 1.091 | 0.913 |
| | GMM | - | 1.731 | 1.215 | 1.056 | 0.856 |

Table 7: **Generalization with Increased Observations.** Error of different models when executed *at test time* on different number of observations across different distributions. We train models with 3 components and 30 observations.

