# OpenReview forum: "Learning Online Data Association"
_TMLR — Withdrawn by Authors_

### Review · Reviewer_UvhH · 2023-03-08

**Summary Of Contributions:**

This paper proposes a method for “dynamic entity monitoring”. Given a sequence of (noisy) observations in the form of object detections, the goal is to determine which individual objects generated each observation and estimating the state of each object based on its observations. A special case of this problem occurs when the objects’ state is static (relative to the observer), in which case this can be treated as an online clustering problem.

The proposed method (DAF-nets) is loosely inspired by the probabilistic data association filter (DAF; Bar-Shalom et al., 2009), which is described as being cumbersome to apply in practice.
The DAF-net works by maintaining a set of K latent representations as its state: one for each object. At each time-step, when a new observation comes in, it is encoded into a latent vector of the same dimension as each of the latent states. In this way, the encoded observation and the latent states can be “matched” via attention to determine which object is expected to belong to (this basically implements an E-step in an EM approach to clustering). Some post-processing is used to “suppress” the resulting attention weights, such that only the top M matches will end up being matched to the encoded observation. For these states, an update is computed based on the encoded observation and the state is updated. This is a fairly sophisticated process, which additionally includes computing a relevance value to mitigate the effect of outliers. Outputs are computed from the updated states using a decoder while the state is further updated using a transition module to capture object dynamics. Confidence scores for each state are derived from the proportion of observations that each state accounts for.

DAF-nets are trained using a 3-part loss in a supervised manner based on the true current state of objects at each timestep in the sequence. The first term ensures that each ground-truth object is well represented by at least one of the estimated states. The second, complimentary, term ensures that each estimated state captures one of the ground-truth object states. Finally, a sparsity loss (which is only activated during the later stages of training) is used to discourage the model from using different states to describe the same ground-truth object.

A brief theoretical analysis is included to motivate the choice of using a single network operating on each individual state, as opposed to having a single network simultaneously updating all states at once. In particular, it is argued that the former choice yields to lower sample complexity. Similarly, it is shown how the sparsity loss used here will lead to sparsity.

DAF-nets are evaluated in an online clustering setting and compared to online and batch (learned) clustering approaches, where it is shown to perform better. In the dynamic entity monitoring setting it is shown how it is competitive to a hand-designed probabilistic DAF method. Finally it is shown how DAF-nets can perform this task for simple images like MNIST images.


**Audience:**

No

**Broader Impact Concerns:**

N/A.

**Claims And Evidence:**

No

**Requested Changes:**

In my view this paper has too many issues to be considered for publication in this shape and form. This is disappointing, since this is a second attempt already (and the second time I invest time) after I previously provided feedback that the paper hadn’t been updated since 2020 (though it now includes 3 citations for the period 2021 - 2022). That aside, there are a number of changes that could be made to make the paper stronger:

* Currently the baselines are trained differently using K slots where this is applicable as opposed to the 10 (or more) used by DAF-Net. I would like to see an experiment where for Slot Attention and Set Transformer the same number of 10 components is used (and a top K selection at inference based on the components that have the lowest loss) since the additional components may also give it additional flexibility, especially when not having access to confidence scores.

* SAVi (Kipf et al., 2022) is an extension of slot attention that includes a dynamics model, and which could be used as a baseline in the dynamic setting.

* It was not clear to me how the confidence scores can be regularized to become sparse. My understanding from Algorithm 1 is that they denote the fraction of observations that have been assigned to a particular slot. Thus why would we ever expect this to become 0 or 1? Having a per-example confidence across slots would make more sense, though in that case one would need a length K confidence vector for each observation, which doesn’t seem to be the case.

* While this paper positions DAF-net for the problem of dynamic entity monitoring, only a single experiment is conducted in this setting and no analysis is provided. For example, a good baseline in Table 5 would be having DAF-Net without the dynamics model that simply replaces the new state with the up date. It would also be interesting to see how the state evolves in the absence of any input to validate that the object dynamics are tracked correctly. Conversely, is the transition module needed for the online clustering experiments?


**Strengths And Weaknesses:**

**Strengths:**

* The strongest comparison in this work is to an online formulation of the Set Transformer, where DAF-nets are shown to compare favorably. At the same time, the Set Transformer is a more general architecture that can be adapted to a variety of settings, while DAF-nets were specifically designed for these types of problems.

* Within the online clustering setting (with dynamics) DAF-nets are shown to be quite versatile, and the slot-based design offers it flexibility to adapt at test time to cases when more objects are present.

**Weaknesses:**


* I think the paper is quite poorly written and it took me quite a while to understand how the proposed method works. While this paper spends a lot of time talking about the similarities to DAF, these are never made explicit. Figure 1 offers a generic comparison in terms of how the object states are updated iteratively, yet for other supposed similarities like the update module, it is unclear how DAF and DAF-nets compare. A step-by-step comparison of the DAF estimation algorithm (either using the fully Bayesian update or the maximum likelihood ML-DAF approach) to DAF-net in Algorithm 1 is crucial. If this connection is indeed important then the reader should not have to first read the 19-page tutorial from 2009 that is currently cited instead of this.

* Critical details about the proposed method and the baselines are missing, to the extent that the results presented here can not be reproduced.

  * About DAF-Nets: How is M determined, which is used for suppression? What is the functional form of the transition module, is this the same as NN3? How are DAF-nets trained, i.e. using what optimizer? For how long is it trained and after how many epochs is the sparsity loss introduced?

  * About the baselines broadly: What does it mean that the baselines are “given and asked to predict the ground-truth number of components K”? Appendix A.3 states that “We use the Chamfer loss to train baseline models, with confidence set to 1”, which is more inline with the reported MSE error. In terms of the input, what does it mean to “give the ground-truth number of components K”? Is this a numeric value provided as an additional input? Are the baselines trained with the first and the second loss term or only the first term?

    * About the LSTM baseline: It is unclear what the output of the LSTM looks like. Is it trained to predict the ground-truth state of all objects simultaneously at each step?

    * About Vector Quantization: No details about any of the hyper-parameters are provided.

    * About the GMM: No details about any of the hyper-parameters are provided.

    * About Slot Attention: No details are provided at all, even though it includes many hyper-params like the choice of initialization, the number of iterations, etc. Since Slot Attention is also a slot-based method, I would expect it to perform quite well, though it is reported to perform poorly. The appendix does not include any mention of the architecture used.

* The proposed method is rather complex and it is unclear why certain design choices are made. For example, the “attend”, “update”, and “relevance” functions operate both on the state, the encoded observation and the length K vector of counts n. Why are the counts included in this part of the computation? The sparsity regularizer is only evaluated in the absence of the learned memory, suppression and relevance, but not when these components are present. It is natural to assume these other components have a bigger effect on performance, which leaves it unclear how much this term actually contributes in the end. Using an MLP to compute attention as opposed to using cross-attention based on cosine similarity, which is highly prevalent these days, is another strange design choice that is not evaluated.

* The paper makes false claims in several places. For example it is written that “[...] none of the \emph{learning} baselines can adapt to different numbers cluster components at inference time, …”, which is not true since Slot Attention can be adapted to using different number of slots at inference time. It is also written that “DAF-Net outperforms non-learning clustering methods, even those that operate in batch mode rather than online, because those methods cannot learn from experience to take advantage of information about the distribution of observations and true object properties (Tables 1, 1 and 6).” However, in Table 1 DAF-Net frequently performs worse to batch mode baselines, especially on “Normal”, “Elongated” and “Mixed”. The claim about the performance of DAF-Net in Table 4 is also a misrepresentation. If the authors wish to claim that DAF-Net is “ [...] performing similarly to K-means++ [...]” while it consistently performs slightly worse then they should not also claim that “DAF-Net [is] out-performing the strong online baseline VQ” when it performs slightly better by the same margin. More generally, it is unreasonable to claim that one method outperforms the other here for such margins, especially when only considering a single seed.

* For the majority of the experiments only a single seed is provided.

---

### Review · Reviewer_XBsz · 2023-03-10

**Summary Of Contributions:**

The paper proposes DAF-net - a neural architecture inspired by data association filters (DAFs) and capable of solving data association problems like online clustering or tracking. The model is evaluated on three domains: 1) online clustering with fairly simple synthetic data, 2) a type of an object tracking problem, with noisy coordinates of a single object as observation at each time step, and true positions of all objects as outputs, 3) a synthetic image-based task with images as inputs and outputs. The method outperforms the provided baselines on all three, especially for longer observation sequences.

**Audience:**

Yes

**Broader Impact Concerns:**

No particular concerns. The method proposed here can be used for questionable purposes (for instance, unethical applications of tracking), but so is basically any technology.

**Claims And Evidence:**

No

**Requested Changes:**

Please address the points made in "weaknesses "above, in particular:
1. More convincing evaluation - or at the very least a clear explanation of why the provided evaluation is sufficient to show the practical value of the proposed method
2. Better tuning of the baselines, in particular Set Transformer
3. Clarify and fix the various presentation issues mentioned in "weaknesses"

**Strengths And Weaknesses:**

Strengths:
1. A reasonable architecture, insth inductive biases inspired by some classical methods
2. Good empirical performance on the provided tasks
3. Mostly good presentation (although see some notes below)

Weaknesses/questions:
1. I found the choice of the tasks used for evaluation somewhat unconvincing - they all come across as somewhat toy/artificial and created anew by the authors, which means there are no prior results reported on them. While it is great that the proposed method works well on these tasks, how much do these results really tell us about the performance of the proposed method on practical problems it is intended to solve? Would it be possible to directly evaluate on some practical problems of interest, where established baselines exist? The dynamic task potentially may fall in this category, but it still looks a bit toy/simplistic. And the image task is very artificial - why not do MNIST clustering, which seems like a much more usual/natural task? Or unsupervised object discovery, although not sure if it fits in the DAF formalism. Tracking (vision-based), at least in some toy/simple setup, could be another interesting task.
2. Comparison to Set Transformer seems unfair - the number of slots is set to a smaller number (maybe more slots would help it?) and also the loss used is different - IIUC, one-sided Chamfer loss, while the proposed method uses more or less “symmetric” Chamfer loss additionally weighted using confidences. The Set Transformer baseline should be tuned at least by trying the symmetric chamfer loss and different numbers of slots. Same for Slot Attention, potentially the other baselines too, but I’m less familiar with those (note: as mentioned, I’m not sure how the proposed model handles having more slots than objects, so I don’t know if the same approach is applicable to the baselines, but maybe it is or maybe one could come up with some ad-hoc replacement - like at the end cluster the predicted K hypotheses with k-means or so to the desired number of objects)
3. On dynamic objects, the gap between the proposed method and LSTM is pretty small and could potentially be explained by, say, different amounts of hyperparameter tuning put into the different models. It would be more convincing to see wider gaps - perhaps for longer observation sequences?
4. On dynamic objects, why would Set Transformer not be able to learn to do the task? Can it be a matter of tuning or is there some fundamental limitation that doesn’t allow it to do it? Seems strange that it would fail completely, while being a good general function approximator.
5. In “Performance on More Clusters.”, how exactly are the clusters sampled? Something is said in the caption of Table 4, but I do not understand it. I generally find the setting quite weird - training with 30 clusters and 50 observations, meaning that on average there are fewer than 2 points per cluster? Seems difficult, potentially somewhat ill-posed (depending on the distribution of the observations) and perhaps an unnecessarily large jump compared to the previous tasks.
6. For Proposition 1 there seems to be only a sketch of the proof, I didn’t find the proof in the appendix. This is not ok: if there is a proposition, there should be a complete proof.
  6a. Moreover - although that is a bit of a matter of taste - I think the proposition (or at least its proof) could be moved to the appendix. Its message that MLP is not going to be efficient at learning the task is fairly obvious practically speaking, so it does not seem to add much to the flow of the paper, but it is difficult to understand, since it heavily builds on prior work.
  6b. The proof of Proposition 2 I believe could also safely be moved to the appendix, since it does not add to the flow of the paper, but rather distracts the reader
7. I’m not sure what happens when there are more slots in the model than objects/clusters in the input data. For instance, Figure 1 shows several red dots per cluster. IIUC each red dot corresponds to a “slot” in the model? So a few slots can be assigned to one cluster? And the model does not predict how many clusters there are? How is the error on cluster means (as reported in Table 1) computed, then? The section “Inferring Object Number.” seems related, but it comes later and doesn’t seem to exactly clarify everything. I guess the top hypotheses are selected based on confidence, but it somehow wasn’t clear from the text (maybe I missed it)
8. Smaller presentation questions/issues:
  8a. Beginning of section 3 - do T and L_i have the same meaning? Then why different letters?
  8b.It’s strange that the losses are discussed in the “Problem formulation” section - they seem to be a part of the solution, not the problem
  8c. “Chamfer distance/loss” in computer vision often refers to the “symmetric” version of the distance, see for instance https://graphics.stanford.edu/courses/cs468-17-spring/LectureSlides/L14%20-%203d%20deep%20learning%20on%20point%20cloud%20representation%20(analysis).pdf or https://www.tensorflow.org/graphics/api_docs/python/tfg/nn/loss/chamfer_distance/evaluate (as well as many papers). Perhaps worth pointing out. Also, since the weights derived from confidences are applied, perhaps it should be called “weighted chamfer loss”?
  8d. Encoding step is not illustrated in Figure 1, makes it a bit more difficult to match the figure to the text
  8e. In the formula on top of page 5, defining w_k , it’s not clear what kind of an operation “attend” is - learned/not, deep/not. Will probably be clearer later, but it would be nice to hint already here. Also, typically the whole thing, including the softmax, would be called attention.
  8f. Same about “update”, “decode”, and “transition” - I guess they are neural networks and about “transition” it’s even mentioned, but only somehow in passing - would be great if the paper was clearer on this. (this is mentioned in the appendix, but should be mentioned in the main paper)
  8g. “The a vectors are integrated to obtain n, which is normalized to obtain the output confidence c.” - integrated over what and normalized by what? It’s clear from the algorithm in Figure 1, but it would be nice if the text would be more self-contained too.
  8h. “Because most of the ak values have been set to 0, this results in a sparse update which will ideally concentrate on a single slot to which this observation is being “assigned”.” - I’m a bit confused, does this suggest there’s only one object per observation - that is, the observation can/should be assigned to one slot? I thought that’s not necessarily the case?
  8i. “slot-based architectures enables” (beginning of Section 5)
  8j. “(Tables 1, 1 and 6).” (bullet point list in the beginning of Section 6)
  8k. Figure 3 -> “OBM-Net”?

---

### Review · Reviewer_RHQR · 2023-03-10

**Summary Of Contributions:**

This paper proposes a learnt online clustering algorithm, which leverages a learnt encoder and uses Transformers to match inputs to potential clusters. It uses some ad-hoc algorithmic modifications to make this process work more stably, without requiring a-priori knowledge of the number of clusters. It is evaluated on toy synthetic data and toy image datasets and outperforms appropriately selected baselines.

Overall, this is an interesting paper which tackles a niche but relevant problem. I have some issues with the overall framing of the paper and there are some sections which do not support claims as well as I’d like, but I’d tend towards acceptance already in this state, given TMLR’s reviewing guidelines.


**Audience:**

Yes

**Claims And Evidence:**

No

**Requested Changes:**

1. Remove mentions of “object hypotheses” and unify on using an online clustering nomenclature.
2. Remove/rework section 3 to keep Section 4 as the relevant mathematical formulation.
3. Remove the theoretical analysis or tone done its proposed contribution
4. Clarify model choices and why the algorithm couldn’t be modified to handle removing cluster centers

**Strengths And Weaknesses:**

1. The paper introduces the problem well and assesses existing literature well enough. The actual problem of online clustering is complex, and any learnable solution is valuable. Overall, I feel like the results are trustworthy and demonstrate the validity of the approach, however this is still only on very toy data and I do not know how confident I would be for this method to scale up to complex, multi-object scenes clustering.
2. I found the use of “Object discovery” unhelpful and not strictly correct.
   1. You are doing online clustering, there is no benefit and no evidence from the current results that this is/would correspond to object detection (unless one takes a very specific semantic definition of what “an object” is, which you do not make clear).
   2. This introduces quite a lot of confusion as some sections use “hypothese”, some “slots”, some “objects”, even though really it’s “just” clusters.
3. There are some clarity issues with some of the sections:
   1. Section 3, Problem formulation, was overly confusing to me, especially given it introduces a set of symbols which are directly modified in Section 4. It would have been more helpful to take an actual example throughout (e.g. from Figure 7), and run through it comparatively.
   2. Section 4 spells the model choices quite well, but it would be valuable to stick to one term as explained above (i.e. “slot”, “cluster”), instead of using “objects” or memory in various.
   3. Please write the full loss used in terms of symbols of Section 4, and not by just referring to Section 3.
4. Section 5, Theoretical analysis, does not provide enough evidence to the claim that “DAF-Net may learn to construct an optimal DAF”, and in my opinion can be relegated to the Appendix.
   1. Proposition 1 is assessing sample complexity, which I do not see how this relates to finding “the correct hypothesis assignment after seeing N samples”
   2. Proposition 2 feels trivial and I do not really see what value it adds?
5. Some model choices are questionable and aren’t ablated:
   1. Why do you need s_k to have this particular form? What happens if you directly write to s?
   2. Why do you use r=relevance as this particular extra DeepSet (also, please call it a DeepSet…)? Why didn’t you use w_k directly?
   3. Why do you use n_k as the integral of a_k?
   4. These all feel like they could be simplified further? If that is what the Set Transformer does, please flag differences more precisely, and explain why you came to this formulation.
   5. The transition function feels both overkill in simple situations and not powerful enough when dynamics will actually be necessary (i.e. it is not action-conditioned and hence can only represent self-dynamics).
6. There is no mechanism in the model to “remove” unused cluster centers?
   1. This feels like a clear disadvantage of this method. The confidences would obviously help to prune unused centers, but it feels like the method likes to use and keep too many currently? (Figure 2, 4 and 7).
   2. Figure 6 is too noisy and actually feels like a negative result? At best it shows that the model does not adapt that well to many components?

---

### Comment · Reviewer_UvhH · 2023-03-16
**Case of (self-)plagiarism: Identical method has previously been published elsewhere + parts of text are verbatim copied from elsewhere.**

Today I stumbled across "Learning Object-Based State Estimators for Household Robots" (Du et al., 2022), which was published at IROS and is not cited in the work under consideration. However, the paper under consideration contains many paragraphs that are verbatim copied from the IROS paper. In particular passages of the introduction, the entire method description etc.

Compare also Figure 2 in the Du et al. paper to Figure 1 in the paper under consideration, or Algorithm 1 (OBM-Net execution) in the Du et al. paper to Algorithm 1 (DAF-Net execution) under consideration, which is identical in all regards.

Note that TMLR states that "Unlike many other journals, TMLR only accepts original contributions that don’t reuse the authors’ own prior work. **In particular, we do not accept submissions that are expanded versions of conference papers.** There should not be any reuse of written text, figures or results between the submitted paper and any paper which has been published, accepted for publication, or submitted in parallel at another archival, peer-reviewed venue.". https://jmlr.org/tmlr/editorial-policies.html

Thus the work under review clearly constitutes a case of (self-)plagiarism.

---

> ### Author Response · Authors · 2023-03-17
> **Thanks**
>
> Hi, thanks for letting us know -- we were not aware that TMLR does not allow for extensions of prior work.

---

### Note · Authors · 2023-03-17

**Comment:**

We thank reviewers for their time spent reviewing our paper. We were unaware that TMLR does not allow submissions that build on prior published work and have withdrawn the paper. We apologize for the time wasted by reviewers reviewing the paper.

**Withdrawal Confirmation:**

I have read and agree with the venue's withdrawal policy on behalf of myself and my co-authors.